# Impact of the Electric Vehicle Policies on Environment and Health in the Beijing–Tianjin–Hebei Region

**DOI:** 10.3390/ijerph18020623

**Published:** 2021-01-13

**Authors:** Chenen Ma, Lina Madaniyazi, Yang Xie

**Affiliations:** 1Department of Industrial Engineering and Economics, Tokyo Institute of Technology, W9-74, 1-12-1 Ookayama, Meguro-ku, Tokyo 152-8552, Japan; ma.c.ac@m.titech.ac.jp; 2Institute of Tropical Medicine, Nagasaki University, Nagasaki 852-8523, Japan; 3School of Tropical Medicine and Global Health, Nagasaki University, Nagasaki 852-8523, Japan; 4School of Economics and Management, Beihang University, Beijing 100191, China; 5Beijing Advanced Innovation Centre for Big Data-Based Precision Medicine, Beihang University, Beijing 100191, China; 6Key Laboratory of Big Data-Based Precision Medicine, Ministry of Industry and Information Technology, Beihang University, Beijing 100191, China

**Keywords:** electric vehicle policies, Beijing–Tianjin–Hebei, air pollution, health impact, co-benefit

## Abstract

China is implementing intensive policies on electric vehicles to control air pollution in urban regions, especially the Beijing–Tianjin–Hebei (BTH) region, one of the most polluted areas in China. The development of electric vehicles will lead to an increase in electricity demand. Because electricity is mostly generated by thermal power in China, primary energy consumption will also increase. This study sets two scenarios: with the electric vehicle policies scenario (REN) and without the electric vehicle policies scenario (FOS) to compare electric vehicle policy’s impact. We quantified the health benefits of the electric vehicle policies in the BTH region by using an integrated assessment framework. Compared with scenario FOS, the local PM_2.5_ emission will reduce by 11.38%, 15.12%, 22.27%, and the concentration will reduce by 18.84%, 20.04%, and 19.57% in Beijing, Tianjin, Hebei separately by 2030 in REN. The electric vehicle policies can avoid 23.5 million morbidities and 4.6 thousand mortalities and save CNY 20.65 billion using the value of statistical life and 1.5 million work loss days in 2030 in REN. Our results show that electric vehicle policy can bring a remarkably positive benefit to public health and the economy.

## 1. Introduction

### 1.1. Background

About 90% of the population in the world are breathing polluted air every day [1]. Particles with a 2.5 μm or less (PM_2.5_) are the primary air pollutant in China. The concentration of PM_2.5_ in most cities in China exceeds the recommended value of the WHO air quality guidelines by more than four times [1]. There is still a long way to go for China to reach the WHO recommendation of less than 10 μg/m^3^. At present, transportation has become the main source of air pollution in cities. China has been the largest automobile sales country in the world for more than ten consecutive years. The current market size is far more than 18 million vehicles, which is the peak of the United States’ market size [2]. Additionally, the development potential of the automobile market in China is still huge. Under China’s per capita income level in 2016, the number of car ownership per 1000 people was 130, while the number was 645 in the US (World Bank database, https://data.worldbank.org.cn/). However, the growing demand for automotive energy has significantly affected the environment [3]. To alleviate the environmental pressure brought by the transport sector’s prosperity, China has vigorously implemented electric vehicle policies [4]. However, using electricity has some adverse environmental impacts, since the generation method in China is mainly thermal power, which mainly uses coal. And the number of pollutants produced by coal consumption is huge including PM_2.5_ [5,6]. Additionally, these pollutants will result in the occurrence of various diseases. PM_2.5_ can lead to lung cancer, cardiovascular and cerebrovascular diseases [7], and significantly reduce the efficiency and quantity of labor [8].

The BTH (Beijing–Tianjin–Hebei) region is a densely populated area with concentrated energy-intensive industries and massive vehicle ownership. In 2010, the BTH region had 2.2% of the land area, but about 7.7% of the population in China, and approximately 14.11% of the country’s civil automobiles (National Bureau of Statistics (NBS), http://www.stats.gov.cn/). Once replacement with electric vehicles occurs, PM_2.5_ concentration will decrease in the BTH region. Air quality improvement will have benefits on public health. Electric vehicles can reduce air pollutant emissions, but they need more electricity from the power sector, leading to more fossil fuel consumption for power generation. This study adopts an integrated methodology to evaluate the interaction between the environment, health, and economy of electric vehicle policies in the BTH region.

This study is structured as follows. A short introduction and literature review are described in Section 1. In Section 2, scenario settings, the energy demand projection method, the GAINS (Greenhouse Gas and Air Pollution Interactions and Synergy) model, and the Integrated Model of Energy, Environment and Economy for Sustainable Development/Health (IMED|HEL) are described. Section 3 shows the results of the environmental impact and health impact. Section 4 summarizes the major results and the reasons that may lead to these results, and the paper concludes in Section 5.

### 1.2. Literature Review

A lot of studies have investigated the environmental and health impact under different electric vehicle development policies. Some studies have focused on the environmental impact of electric vehicles (EV). Martina Siskova, et al. [9] found that EV can help reduce local and global emissions in most urban regions. Boqiang Lin, et al. [10] estimated that EV’s lowest average environmental values are CNY 30.60 thousand in the four most significant cities in China. Some of them, on the other hand, focus on the health impact of EV. Wen-Y.L. et al. [11] found that the health benefits of EV would be USD 43.35 billion in northern Taiwan, using Weather Research and Forecasting model (WRF), Community Multiscale Air Quality model (CMAQ), and Benefits Mapping and Analysis Program (BenMAP). Shuai P., et al. [12] used a high-resolution WRF-SMOKE-CMAQ-BenMAP air quality and health modeling framework and found that EV could prevent 114–246 premature deaths in the Greater Houston Area in 2040. Yijun G., et al. [13] used an integrated assessment model to study the Greater Toronto and Hamilton Area and found that with 25% and 100% EV penetration rates, EV could reduce about 50 to 260 premature deaths per year. However, few studies have focused on the health impact of electric vehicles in China. Xinyu L., et al. [14] showed that by electrifying 27% of private vehicles, the number of annual premature deaths nationwide could be reduced by 17,456 in 2030 using the WRF model, CMAQ model, and concentration–response relationships. Wei P., et al. [15] found that EV could bring the greatest benefits to the North China Plain region in winter.

Previous evidence in China has mainly been gathered at the county level, with limited information on the regional scale, e.g., BTH, one of China’s most populated and polluted regions. More importantly, previous assessments were based on policy assumptions, however, this study uses empirical data and published policies to assess whether the existing electric vehicle policies provide significant benefits to the BTH region’s public health. This paper uses the GAINS model to simulate PM_2.5_ concentrations and uses the IMED|HEL model to assess the health impact. Both models are widely used. Xu T., et al. [16] use GAINS model and IMED|HEL model to evaluate PM_2.5_ pollution-related health effects in China’s road transport sector. Xiang Z., et al. [17] and Shaohui Z., et al. [18] use GAINS model and IMED|HEL model to evaluate the health benefit of the residential heating and cement industry in the BTH region.

## 2. Methods

This study uses an integrated assessment approach to investigate the environmental and health impacts of electric vehicle policies in the BTH region. Our analyses were completed in three steps (Figure 1). Firstly, we mainly used the least square method to fit polynomial functions to predict vehicle data and energy consumption data. Secondly, we used the GAINS (Greenhouse Gas and Air Pollution Interactions and Synergy) model to assess the pollutant emissions and the annual average PM_2.5_ concentration in the BTH region. Finally, we used the IMED|HEL model to evaluate the health impacts of PM_2.5_ exposure, including premature deaths, morbidity, work time loss, and economic loss.

### 2.1. Scenario Setting

This study establishes two scenarios, a scenario with the electric vehicle policies (REN) and a scenario without the electric vehicle policies (FOS) (Table 1). The ownership of electric vehicles in China was only 12.8 thousand in 2012, and there was no supporting policy, and the technique was immature [2]. Therefore, we assume the base year to be 2010, and that there were no electric vehicles in the BTH region in 2010. We assume all vehicles to be traditional-fuel-consumption vehicles (FVs)—both currently and in the future—under the FOS scenario. The development of electric vehicles heavily relies on policy support, and its development in China started in 2010 [19]. Under the REN scenario, electric vehicles are assumed to develop rapidly based on supportive policies. We consider both the local generation and input of the electricity consumed in the BTH region. This study mainly considers two types of vehicles, pure electric vehicles (EV) and plug-in hybrid electric vehicles (PHEV), which are promoted more in China [4]. In this study, we use EVs to represent electric vehicles, including EV and PHEV.

### 2.2. Energy Calculation Method

#### 2.2.1. Vehicle Ownership Forecast

Since energy consumption varied from different types of vehicles, this study divided traditional fuel consumption vehicles into five categories. Small and micro passenger cars were classified as passenger vehicles (PC). Trucks with a total weight of 6 to 14 tons were classified as medium-duty trucks (MT), heavy-duty trucks (HT) were classed as having a total value of more than 14 tons, and large passenger cars (HB) as having a total length of more than 10 m. All other vehicles were classified as light commercial vehicles (LC). We used automobile national prediction data towards 2030 from [20]. Due to supportive policies [21], we first needed to satisfy the total ownership of the BTH region in the target year and then to satisfy the proportion of different types of vehicles. Therefore, based on the historical data (2001–2017) of China and the BTH region [22], we first calculated the national proportion $r_{c}$ of Beijing, Tianjin, and Hebei separately; secondly, we calculated the proportion of the five types of vehicles $r_{k}$ of the BTH region separately (Table 2). Then we used Equation (1) to predict vehicle ownership. Appendix A shows the vehicle ownership.

We then projected the ownership of EVs (EV and PHEV). Many studies in China separate EVs into three categories, passenger cars, buses, and trucks. Different types of EVs would have different levels of energy consumption. Due to data availability, we only collected the national proportion of EVs in the total vehicle category [23], as shown in Appendix A. Thus, we reclassified the national vehicles [20] into passenger cars, buses, and trucks, to obtain the national ownership of each type of EVs. This study introduced the concept of “electric vehicle city index” and “electric vehicle promotion index” to calculate the local ownership. The indexes included the market and industrial environment, purchase policy and convenience of equipment, and could better reflect the distribution of China’s electric vehicle ownership and development potential by province. This study integrated the two indices, 0.4 and 0.6, to set the city share function of EVs ownership. Therefore, we used Equation (2) to predict the ownership of EVs. The EVs ownership prediction is presented in Appendix A.
(1)$$y_{ck}^{t}=N^{t}r_{c}^{t}r_{k}^{t}$$
(2)$$y_{ck^{,}}^{t}=y_{k^{,}}^{t}p_{ik^{,}}^{t}I_{c}$$
where:
y indicates the ownership of automobiles, (1) is all automobiles, (2) is EVsN indicates the total national ownershipt represents the year from 2001 to 2030c represents one of Beijing, Tianjin, Hebeik represents a certain type of automobiles (five types: PC, LC, MT, HT, HB)$k^{,}$ represents a certain type of automobiles (three types: passenger cars, bus, and trucks)i represents EV or PHEVr indicates the proportion from Table 2p indicates the proportion from Appendix A (of EVs)I indicates the city share value

#### 2.2.2. Energy Consumption Forecast

We first calculated average energy consumption data (fuel consumption, unit: L/100 km; electricity consumption, unit: kWh/100 km) for each kind of vehicle. About 50 thousand types of vehicles (PC and LC) were investigated on the China Automobile Fuel Consumption Inquiry System (http://www.miit.gov.cn/asopCmsSearch/). The data of HB, MT, and HT were obtained from [24,25]. We combined the future target from official documents [19,26] to obtain the expected fuel consumption data of FVs. Additionally, this study collected the data of electric automobiles from the government documents (http://www.miit.gov.cn/index.html). According to the official document policy objectives, EV’s power consumption includes electricity consumption and fuel consumption, as shown in Appendix A.

Since in the BTH region electricity consumption is far higher than the production, a large amount of electricity is input from other provinces every year. The electricity used by electric vehicles in the BTH region mainly comes from the Jing-Jin-Tang power grid. The main power input sources are the Inner Mongolia province and Shanxi province [27]. Due to data availability, this study used the national cross-regional power transmission situation in 2006. The proportion of power input sources consists of 26.7% of Shanxi’s electricity and 73.3% of Inner Mongolia’s [28]. To calculate the coal consumption, this study used data from the National Bureau of Statistics (NBS) to regress using ordinary least squares (OLS) (R^2^ ≥ 0.9). The prediction functions are in the form of Equation (3) and the prediction results are shown in Appendix A.
(3)$$y_{c}=\alpha +\beta x$$
where:
y indicates the energy, the unit is Billion kWh.x indicates the year from 2001 to 2030(α,β) are coefficients of the function, as shown in (Table 3).

#### 2.2.3. Energy Calculation

The two scenarios’ energy consumptions were calculated separately, considering the annual mileage and the cross-province transmission line loss rate [29]. In the FOS scenario, energy consumption was limited to fuel consumption, and the calculation method is shown in Equation (4). The energy consumption in the REN scenario was divided into electricity consumption and fuel consumption. As for the EVs, the calculation method is shown in Equation (5). To obtain the fossil fuel consumption in the REN, electric power consumption was converted into coal consumption from thermal power plants in five regions (including Shanxi and Inner Mongolia provinces). The method is shown in Equation (6). In this study, the calculated data were converted into the energy index. The calorific value of standard coal is about 29,270 J/g, and gasoline is about 44,000 J/g, while the density of gasoline is about 0.75 g/mL. By combining the information above, we can obtain the energy consumption of the two scenarios (Appendix A).
(4)ℂfcfv,t=∑k=1noc,kfv, tpfkfv,tmkfv, t
(5){ℂect=∑k′=1m(oc,k′ev, tpek′ev,tmk′ev, t+oc,k′ph, tpek′ph,tmk′ph, tRk′)ℂfcph,t=∑k′=1moc,k′ph, tpfk′ph,tmk′ph, t(1−Rk′)
(6){ ℂcoct=ξ(TctGct)ℂect(GctCct)ℂcoc′t=ξ (Tc′tGc′t)tc′∑c=13ℂectCct−GctCct(1−λ)
where:
ℂf, ℂe, ℂco represent fuel, electricity, coal consumption of automobiles, respectively.fv,ev,ph represent FVs, EV, and PHEV, respectively.c, c′, t represent the provinces in the BTH region, province of Shanxi or Inner Mongolia, and which year, respectively.o, m, pe,pf represent ownership, annual mileage, electricity, and fuel consumption per 100 km, respectively.n, m represent the total number of FVs types, and the total number of EVs types, respectively.k represents a certain type of automobiles (five types: PC, LC, MT, HT, HB)k′ represents a certain type of automobiles (three types: passenger cars, bus, and trucks)R represents the ratio of PHEV using electricity in 100 km.G,C,T represent electricity generation, electricity consumption, and thermal power generation, respectively.ξ,λ,t, indicate how many grams of standard coal is consumed by 1 kWh of thermal power generation, the loss rate of the transmission line, and transmission ratio, respectively.

### 2.3. GAINS Model

The Greenhouse Gas and Air Pollution Synergy (GAINS) model was developed by the International Institute for Applied System Analysis (IIASA) in Austria. The introduction documents, the principal data, assumptions, and results are freely available online (IIASA website: https://gains.iiasa.ac.at/gains/, accessed 24 October 2020). The GAINS model estimates future potential emission reductions of almost 10 air pollutants and six greenhouse gases (GHGs) for each country and costs based on approximately 2000 specific emission control measures from countries or regions all over the world [30]. The model is used to assess the environmental impact of human social activities. It can calculate the reduction in certain pollutants or GHGs and air pollutant concentrations based on certain areas’ policies [31]. Many studies employ this model to explore environmental impacts [17,32,33]. The basic principle of calculating emissions in this model (in a stylized way) is Emissions = Activity * Emission factor * Technology implementation. This paper mainly changes the energy activity data and mobile source in two scenarios (using the energy consumption data from 2.2) and the pollutants emissions and the PM_2.5_ concentration are the two main goals [34,35,36,37,38].

### 2.4. IMED|HEL Model

The IMED|HEL can quantify the health damage caused by air pollution and the monetary value of health impacts. The IMED models are continuously updated and the documents of detailed information on the model are available online (http://scholar.pku.edu.cn/hanchengdai/imedhel). Exposure to incremental PM_2.5_ concentrations leads to several health endpoints. In this study, we adopted the linear concentration–response functions (CRF) from [33,39] to calculate the morbidity of several diseases, premature deaths, and work loss days. The linear function assumes the CRF is a constant and in the 95% confidence interval [40,41,42]. We mainly used the medium value of the CRF. In this paper, the work loss days (WLDs) include both the premature deaths between 15–65 years old and morbidity for labor. This study used the population data from the sixth census to conduct the calculation. Additionally, we monetized the health impact, especially the economic loss of premature deaths, using the value of statistical life (VSL) (Unit: million CNY in 2002 price), which was calculated by certain province’s current GDP per capita values relative to the national average per capita GDP in 2010 and income elasticity of 0.5 [43,44].

## 3. Results

### 3.1. Prediction of Energy Consumption

The results show that the total vehicle ownership will peak before 2025 in Beijing and Tianjin. The number of electric vehicles will increase exponentially. In Beijing and Tianjin, EV and PHEV passenger cars and PHEV trucks will have larger ownership (as shown in Appendix A). By 2030, the largest share of electric vehicles in the BTH region will be EV passenger cars. Mainly EV buses and PHEV trucks will increase in the future (in Appendix A). This study estimates fossil energy consumption in the road transport sector. Figure 2 shows the fossil energy consumption in two scenarios. Fossil energy consumption in the FOS scenario is higher than in the REN Scenario. In the BTH region, in 2015, the REN scenario saves 14% of the fossil energy compared to the FOS scenario, while it will save 33% of fossil energy in 2030 (in Appendix A). The fossil energy reduction is significant in Hebei province. Electric vehicles can save about 18% of fossil energy consumption in 2015 and will save about 37% in 2030 in the road transport sector.

### 3.2. Environment Co-Benefit

The GAINS model simulates the emission and concentration of pollutants. The EV policies lead to the local emissions of SO_2_ being significantly reduced, followed by PM_2.5_. By 2030, SO_2_ and PM_2.5_ emissions will reduce by about 19% and 11%, respectively, in Beijing, and by about 35% and 22% in Hebei, compared to the FOS scenario. Table 4 shows that Hebei has the most extensive local emissions reduction, and Beijing has the smallest compared with the FOS scenario. In 2030, the PM_2.5_ emission reduction is about 22% in Hebei, about double of Beijing. PM_2.5_ and SO_2_ emission reductions will increase, while CO_2_ will decrease slightly in this region. The decline in CO_2_ emissions is most obvious in Tianjin, while there is almost no decrease in Beijing.

Figure 3 (annual PM_2.5_ concentration distribution in two scenarios) shows the PM_2.5_ concentration in the BTH (Beijing–Tianjin–Hebei) region. This study uses the concentration data (a grid data of one longitude and one latitude) from the GAINS model. In both scenarios, the southwest of the BTH region is the most seriously polluted, including Xingtai City, while the northern part of the BTH has the lowest PM_2.5_ concentrations, for example, Chengde City. PM_2.5_ concentrations in Beijing and Tianjin are not as high as in Hebei, but the PM_2.5_ concentration still reaches 78 μg/m_3_ in 2015 in some cities in the FOS scenario (in Table 5). EV policies will reduce PM_2.5_ concentration, especially in the over polluted areas. PM_2.5_ concentration reduction will be about 18.8%, 20%, 19.6% in Beijing, Tianjin, Hebei, respectively, in 2030. EVs will reduce PM_2.5_ concentration by approximately 8.71 μg/m^3^, 10.99 μg/m^3^, 8.63 μg/m^3^, respectively. The most polluted areas will have the largest decline, such as Beijing 12.12 μg/m^3^ (59.84 μg/m^3^–47.72 μg/m^3^), Tianjin 12.69 μg/m^3^ (61.29 μg/m^3^–48.60 μg/m^3^) and Hebei 16.14 μg/m^3^ (81.36 μg/m^3^–65.22 μg/m^3^) in 2030. The reduction rate will increase in the long-term. For example, the average reduction rate is 10.28% in 2015 and 18.84% in 2030 in Beijing, and 10.75% in 2015 and 19.57% in 2030 in Hebei.

### 3.3. Health Assessment

This study derives the morbidity, premature deaths, and work loss days caused by the concentration of PM_2.5_. Respiratory symptom cases are predicted to be the highest (about 99% of total morbidity), and the Hebei province contributes about 65% of the total morbidity. In comparison, Beijing contributes about 19% in both scenarios (Figure 4). The electric vehicle policies have the potential to reduce the risk of getting sick significantly. About 23.5 million (20.29~39.03 million, C.I 95%) people can be prevented from sickness in 2030 in this region, including nearly 19.53 thousand (8.31~28.45 thousand, C.I 95%) hospitalization, through the implementation of EV policies.

In Table 6, the premature deaths will decrease in the time dimension, but EV policies can speed up this process. For instance, in the FOS scenario, there would be 24.4 (8~49, 95% C.I) thousand premature deaths in 2015 and 18.5 (7~37, 95% C.I) thousand premature deaths in 2030 in the BTH (Beijing–Tianjin–Hebei) region. However, EV policies can lead to the avoidance of around 3.2 (1.1~6.3, 95% C.I; (2.44–2.13)*10) thousand premature deaths in 2015 and 4.6 (1.5~9.3, 95% C.I; (1.85–1.39)*10) thousand in 2030, in the BTH region. Hebei province has the largest number of premature deaths, but it can also benefit more from the EV policies avoiding more premature deaths. Hebei can avoid 3.5 (1.2~7, 95% C.I; (1.38–1.03)*10) thousand premature deaths in 2030 while Beijing can only avoid 0.6 (0.2~1.2, 95% C.I; (0.25–0.19)*10) thousand and Tianjin can only avoid 0.6 (0.2~1.1, 95% C.I; (0.22–0.16)*10) thousand in 2030. Besides, the reduction rate compared with the FOS scenario in Hebei and Tianjin will be slightly higher than in Beijing. The reduction rate will also increase as time goes. The reduction rate is about 12% in 2015, while it will be about 24% in 2030 in Beijing.

Furthermore, if we consider the monetary value of premature deaths (in Table 7), this region will save about CNY 20.15 (6.7~40.3, 95% C.I) billion in 2030. Hebei province will experience the greatest savings at around CNY 6.22 (2.1~12.4, 95% C.I) billion in 2015 and CNY 13.38 (4.5~26.8, 95% C.I) billion in 2030. Meanwhile, Beijing will save CNY 3.78 (1.3~7.6, 95% C.I) billion, and Tianjin will save CNY 2.99 (1.0~5.9, 95% C.I) billion in 2030 due to EVs policies. The savings increase in the time dimension. The total saving in this region is about CNY 9.46 (3.2~18.9, 95% C.I) billion in 2015, while it will be about CNY 20.15 (6.7~40.3, 95% C.I) billion in 2030.

EV policies will effectively reduce work-loss days (WLD) (Table 8). The total reduction in WLD will be 1.5 × 10^7^ (1.27 × 10^7^~1.73 × 10^7^, 95% C.I) days in 2030 because of EV policies. Hebei province experiences about 63% of the labor loss and is slightly more sensitive to the policies. The reduction rate will reach around 25% in 2030, compared to the FOS scenario in Hebei, while it will be about 24% in the other two areas. Overall, WLD will gradually decrease with time in both scenarios, but EV policies can accelerate this process. In the FOS scenario, the difference in labor loss between 2015 and 2030 in this region is 1.93 × 10^7^ (1.64 × 10^7^~2.23 × 10^7^, 95% C.I) days, but in the REN scenario, it is 2.41 × 10^7^ (2.05 × 10^7^~2.78 × 10^7^, 95% C.I) days. Additionally, the reduction in WLD increases as time goes by. There are about 2.01 × 10^6^ (1.7 × 10^6^~2.3 × 10^6^, 95% C.I) work loss days that can be saved in 2015 while about 2.93 × 10^6^ (2.5 × 10^6^~3.4 × 10^6^, 95% C.I) work loss days in 2030 in Beijing, compared with the FOS. The loss of economic value will be large in 2030 in the FOS scenario, equivalent to the value created by about 0.23 (0.19~0.26, 95% C.I) million laborers in one year (260 work loss days per year). However, in the REN scenario, the WLD of the BTH region will be equivalent to the economic value created by 0.17 (0.14~0.20, 95% C.I) million laborers in one year.

## 4. Discussion

### 4.1. Environmental and Health Co-Benefit

The results show that electric vehicle policies can bring significant benefits to the environment and public health. The carbon emissions of electric automobiles may not decrease when calculated by the full life circle method because electrification needs electricity from coal power plants. The electricity demand increases will lead to increases in carbon emissions [45,46,47]. Still, the policies can reduce pollutant emissions in urban areas remarkably, particularly PM_2.5_ and SO_2_. Hebei province will experience the most considerable reduction in local emissions and the PM_2.5_ concentration compared with the FOS scenario because the current vehicle ownership is low, and EVs will account for a large proportion of vehicle growth. However, Beijing will have the smallest reduction since the current ownership levels are so high that the future increment cannot lead to a significant impact. The northern part of the BTH region has fewer power plants. Most of the power plants in the BTH region are located in the southern part [48]. Simultaneously, population density is also high in these regions. Therefore, the health benefit will be more significant when the proportion of EVs increases. Areas with high PM_2.5_ concentrations are always located around emission sources, such as power plants or urban areas, and are therefore more sensitive to the EV policies [49,50].

Our results show that Hebei province has the highest morbidity levels, premature deaths, and WLD due to Hebei’s large population and higher PM_2.5_ concentration. Therefore, more people will benefit from the improvement in the environment in Hebei. Although the avoided morbidity and mortality levels are lower in Beijing and Tianjin, the economic benefits of implementing EV policies in Beijing and Tianjin are quite significant. Our estimation is consistent with previous works [17,18,51]. Furthermore, the labor loss of Hebei province is slightly more sensitive to environmental improvement, which could be because of the poorer medical care conditions in this region. The PM_2.5_ concentration will continuously reduce, and health benefits will continually increase in the BTH region as time goes by, even without EV policies. However, EV policies can accelerate this process by reducing the usage of fossil fuels. Our result is consistent with previous works conducted in China [14,15,16].

### 4.2. Limitations

There are some limitations in the design of this study. Firstly, because of data availability, there remains some uncertainty about the ownership of EVs. However, we strictly followed the policies, and the predictions can reflect trends relatively accurately. Secondly, we do not consider policies on power generation. The proportion of thermal power generation is expected to reduce, and cleaner sources will develop, such as wind and nuclear. However, thermal power generation replacement will not be that fast in this short period (to 2030). Therefore, we may have underestimated the environmental impact of EV policies. Thirdly, we used the historical population and natural death rate in the health impact assessment and assumed this to be constant until 2030. However, in the future, China will become an aging society; the proportion of older adults will increase. Therefore, this may underestimate the health benefits of EV policies. Finally, this study used an air pollution control strategy in the GAINS model, reflecting the policy implications in 2015, which is behind the technology improvement in China. The co-benefit of the environment and health will be higher than our estimations.

## 5. Conclusions

Improving air quality is an urgent requirement in the BTH region of China. This study uses empirical data and policies to assess whether the existing electric vehicle policies have significant health benefits in the BTH region. This region has a higher population density and the highest vehicle ownership of any region in China. Many pollutants are emitted in the road transport sector and significantly influence public health, bringing economic losses [16]. Electric vehicles can reduce pollutant emissions in the transportation sector. But reducing pollutant emissions—calculated by the full-lifecycle—depends on the energy structure [45,46,47]. Replacing traditional-fuel-consumption vehicles with electric vehicles is necessary, and electric vehicle policies have been vigorously implemented in China. This study quantifies the environmental and health impact of implementing electric vehicle policies in the BTH region using an integrated assessment model. We first calculated energy consumption in the road transport sector from 2010 to 2030 under two scenarios. Then, we input the energy consumption data into GAINS-China to estimate the environmental impact. Finally, we used the IMED|HEL model to analyze the health impact. The outcome of this study is excellent.

This study finds that the implementation of electric automobile policies can effectively reduce fossil energy consumption, thereby improving air quality. It can reduce fossil consumption by 33% in 2030 compared to no policy scenario. In the BTH region, the local emission of various harmful gases is predicted to drop significantly. For example, the local emission of PM_2.5_ will reduce by 11.38%, 15.12%, 22.27% in Beijing, Tianjin, and Hebei, respectively, by 2030 compared with no policy support. PM_2.5_ concentration in the BTH region will decline. By 2030, the average PM_2.5_ concentration is predicted to reach 37.54 μg/m^3^, 43.84 μg/m^3^, and 35.48 μg/m^3^ in Beijing, Tianjin, and Hebei, respectively. Furthermore, the electric vehicle policies have the potential to save 23.5 (20.29~39.03 million, C.I 95%) million morbidities and 4.6 (1.5~9.3, 95% C.I) thousand premature deaths. If we monetarize the value of premature deaths, this equates to CNY 3.78 (1.3~7.6, 95% C.I), 2.99 (1.0~5.9, 95% C.I), 13.38 (4.5~26.8, 95% C.I) billion being saved in Beijing, Tianjin, and Hebei, respectively. The reduction in work loss days will be 24.91% in 2030 when conducting the policies, saving about 1.5 × 10^7^ (1.27 × 10^7^~1.73 × 10^7^, 95% C.I) work loss days. The electric vehicle policies will have significant benefits on the environment and public health in the BTH region. EV policy extension to the other areas will bring more health co-benefits for both the BTH region and others. China should try to improve the proportion of electric automobiles and reach the goal set in official documents. The higher the ratio, the greater the benefit to the environment and health within the BTH region.

## Figures and Tables

**Figure 1 ijerph-18-00623-f001:**
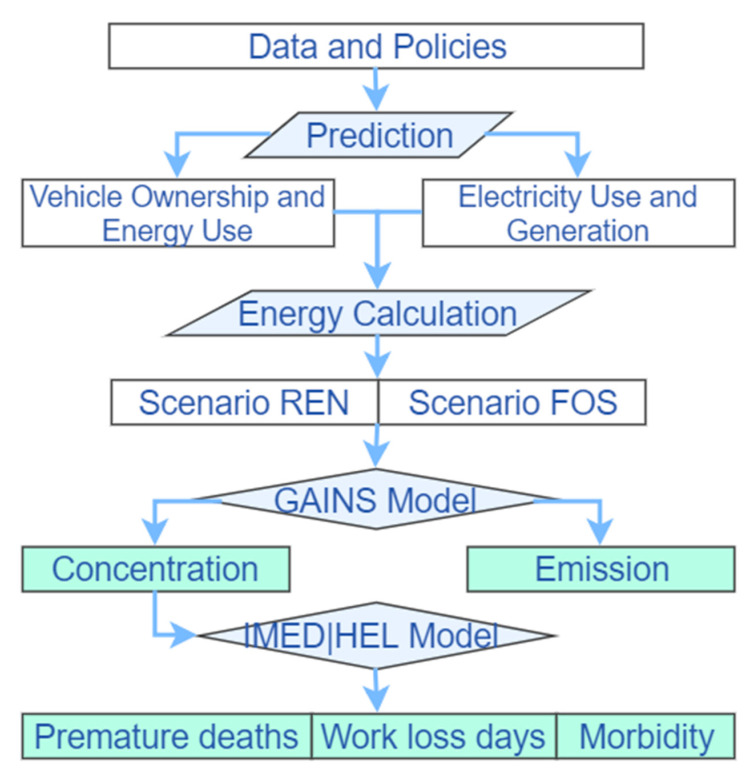
Research framework. REN: with electric vehicle policies scenario; FOS: without electric vehicle policies scenario. GAINS: Greenhouse Gas and Air Pollution Interactions and Synergy; IMED|HEL: Integrated Model of Energy, Environment and Economy for Sustainable Development/Health.

**Figure 2 ijerph-18-00623-f002:**
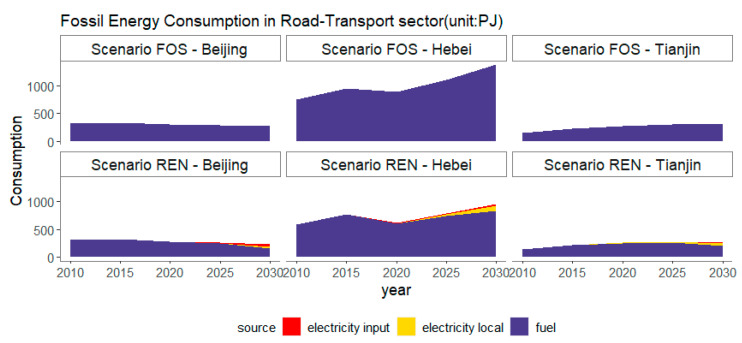
Fossil energy consumption in two scenarios in the BTH region.

**Figure 3 ijerph-18-00623-f003:**
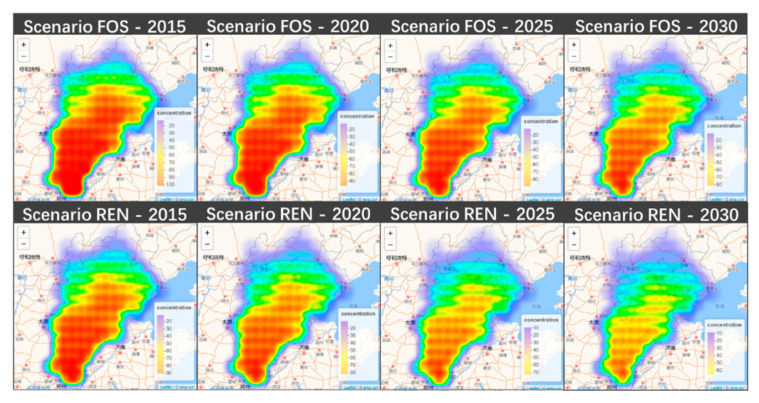
Annual PM_2.5_ concentration distribution in two scenarios.

**Figure 4 ijerph-18-00623-f004:**
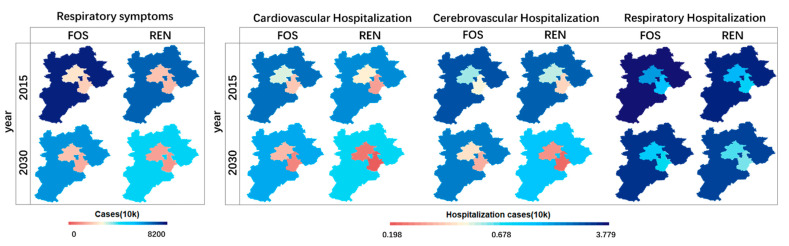
Morbidity in different scenarios.

**Table 1 ijerph-18-00623-t001:** Scenarios setting.

Scenario	The Electric Vehicle Policies	Initial Ownership of Civil Automobiles in BTH Region (2010)
FOS	Not considered	11.01 million
REN	Considered	11.01 million

BTH: Beijing–Tianjin–Hebei.

**Table 2 ijerph-18-00623-t002:** Proportion prediction functions ^1^.

Proportion	Region		
	Beijing	Tianjin	Hebei
$r_{c}$	$8.5\times 10^{-2}e^{-0.058x}$	$-5\times 10^{-4}x+0.0249$	$6.5\times 10^{-2}+rand\left(x\right)$
$r_{k=PC}$	$7.82\times 10^{-2}\ln x+0.7065$	$-7\times 10^{-4}x^{2}+2.8\times 10^{-3}x+0.6185$	0.1903lnx+0.304
$r_{k=LC}$	$-5.8\times 10^{-2}\ln x+0.2157$	$-5\times 10^{-5}x^{3}+2.8\times 10^{-3}x^{2}-5.12\times 10^{-2}x+0.3864$	$0.3658e^{-0.087x}$
$r_{k=HB}$	$-6\times 10^{-3}\ln x+2.65\times 10^{-2}$	$7\times 10^{-5}x^{2}-2.6\times 10^{-3}x+3.18\times 10^{-2}$	$-8\times 10^{-3}\ln x+2.74\times 10^{-2}$
$r_{k=HT}$	$10^{-4}x^{2}-2.7\times 10^{-3}x+2.26\times 10^{-2}$	2$\times 10^{-4}x^{2}-4.3\times 10^{-3}x+4.51\times 10^{-3}$	$2\times 10^{-4}x^{2}-7.8\times 10^{-3}x+8.74\times 10^{-2}$
$r_{k=MT}$	$2.86\times 10^{-2}e^{-0.112x}$	$6.56\times 10^{-2}e^{-0.175x}$	$2\times 10^{-4}x^{2}-1.01\times 10^{-2}x+0.1538$

^1^ This paper mainly uses the least square method to fit polynomial functions ($R^{2}\geq 0.7$). However, the proportion of Hebei province does not have an obvious tendency. After autocorrelation analysis, we found this was a random time sequence with a 92% probability. ($r_{c}$ refers to the proportion of national ownership of each area, $r_{k}$ refers to the proportion of different types of vehicles, x represents the year from 2001 to 2030, with a range of [1, 30]; rand(x) is a random number with a normal distribution of values (−0.003, 0.003)).

**Table 3 ijerph-18-00623-t003:** Coefficient table of energy-related prediction functions (α,β).

Region	Electricity Consumption	Electricity Generation	Thermal Power Generation
Beijing	(−80,728.56, 40.55)	(−29,063.80, 14.61)	(−28,946.59, 14.54)
Tianjin	(−76,252.23, 38.24)	(−60,498.13, 30.35)	(−59,007.31, 29.61)
Hebei	(−33,7849.97, 169.32)	(−238,314.42, 119.56)	(−211,897.89, 106.36)

**Table 4 ijerph-18-00623-t004:** Sensitivity analysis of pollutant reduction.

Region	GHG and Pollutant	Reduction Ratio (%)—Scenario REN Compared with Scenario FOS			
		2015	2020	2025	2030
Beijing	PM_2.5_	5.41	11.74	12.38	11.38
	SO_2_	7.01	9.50	14.77	18.93
	CO_2_	0.00	0.00	0.00	0.00
Tianjin	PM_2.5_	9.96	13.52	14.41	15.12
	SO_2_	13.91	17.73	24.19	29.18
	CO_2_	4.22	3.75	3.29	2.93
Hebei	PM_2.5_	10.86	15.65	19.22	22.27
	SO_2_	14.47	20.07	28.10	35.00
	CO_2_	0.31	0.30	0.27	0.24

**Table 5 ijerph-18-00623-t005:** Annual average PM_2.5_ concentration (μg/m^3^) ^1^.

Region	Year	Concentration in FOS			Concentration in REN			Reduction comparing with FOS (%)		
		Low	Average	High	Low	Average	High	Low	Average	High
Beijing	2015	42.98	58.29	76.65	38.80	52.30	68.45	9.72	10.28	10.69
	2020	40.71	54.92	71.82	35.23	47.03	61.01	13.47	14.36	15.04
	2025	38.00	50.94	66.24	31.91	42.15	54.14	16.03	17.26	18.27
	2030	34.70	46.25	59.84	28.72	37.54	47.72	17.23	18.84	20.25
Tianjin	2015	63.28	69.34	78.06	56.64	61.97	69.59	10.49	10.62	10.85
	2020	59.38	65.05	73.01	50.76	55.48	62.03	14.51	14.71	15.05
	2025	55.05	60.27	67.46	45.29	49.44	55.02	17.73	17.98	18.45
	2030	50.08	54.83	61.29	40.19	43.84	48.60	19.75	20.04	20.72
Hebei	2015	12.85	54.95	101.18	11.54	49.04	90.22	10.16	10.75	10.84
	2020	12.23	51.97	95.92	10.56	44.26	81.48	13.71	14.84	15.05
	2025	11.47	48.34	89.11	9.60	39.71	73.02	16.25	17.85	18.06
	2030	10.53	44.11	81.36	8.70	35.48	65.22	17.39	19.57	19.84

^1^ In each province, there are many concentration points (grid data of one longitude and one latitude), we use the point with the largest value as “high” (most polluted area), and the “average” value is calculated by all concentration points in this province.

**Table 6 ijerph-18-00623-t006:** Premature deaths and morbidity in the BTH region ^1^.

Endpoint	Year	Beijing		Tianjin		Hebei		Total	
		FOS	REN	FOS	REN	FOS	REN	FOS	REN
Premature deaths (unit:10 k)	2015	0.34 (0.1, 0.7)	0.29 (0.1, 0.6)	0.29 (0.1, 0.6)	0.25 (0.1, 0.5)	1.82 (0.6, 3.6)	1.58 (0.5, 3.2)	2.44 (0.8, 4.9)	2.13 (0.7, 4.3)
	2020	0.31 (0.1, 0.6)	0.26 (0.1, 0.5)	0.27 (0.1, 0.5)	0.22 (0.1, 0.4)	1.7 (0.6, 3.4)	1.39 (0.5, 2.8)	2.28 (0.8, 4.6)	1.86 (0.6, 3.7)
	2025	0.28 (0.1, 0.6)	0.22 (0.1, 0.5)	0.24 (0.1, 0.5)	0.19 (0.1, 0.4)	1.55 (0.5, 3.1)	1.2 (0.4, 2.4)	2.08 (0.7, 4.2)	1.62 (0.5, 3.2)
	2030	0.25 (0.1, 0.5)	0.19 (0.1, 0.4)	0.22 (0.1, 0.4)	0.16 (0.1, 0.3)	1.38 (0.5, 2.8)	1.03 (0.3, 2.1)	1.85 (0.6, 3.7)	1.39 (0.5, 2.8)
Morbidity (unit:10 million)	2015	2.39 (2.1, 3.9)	2.09 (1.8, 3.5)	1.93 (1.7, 3.2)	1.69 (1.5, 2.8)	8.13 (7.0, 13.5)	7.04 (6.1, 11.7)	12.45 (10.8, 20.7)	10.82 (9.4, 18.0)
	2020	2.22 (1.9, 3.7)	1.83 (1.6, 3.0)	1.79 (1.6, 2.9)	1.48 (1.3, 2.5)	7.6 (6.6, 12.6)	6.18 (5.4, 10.3)	11.61 (10.0, 19.3)	9.49 (8.2, 15.8)
	2025	2.02 (1.8, 3.4)	1.59 (1.4, 2.6)	1.64 (1.4, 2.7)	1.29 (1.1, 2.1)	6.94 (5.9, 11.5)	5.36 (4.6, 8.9)	10.60 (9.2, 17.6)	8.23 (7.1, 13.7)
	2030	1.79 (1.6, 2.9)	1.36 (1.2, 2.3)	1.46 (1.3, 2.4)	1.1 (0.9, 1.8)	6.17 (5.3, 10.3)	4.59 (3.9, 7.7)	9.42 (8.1, 15.7)	7.06 (6.1, 11.8)

^1^ The values in brackets means confident interval (C.I) of 95%. (C.I low 95%, C.I high 95%), the value out of the brackets means the middle value.

**Table 7 ijerph-18-00623-t007:** The monetary value of premature deaths (Billion CNY in 2002 price) ^1^.

Region	Beijing			Tianjin			Hebei			Total Saved
Year	FOS	REN	Saved	FOS	REN	Saved	FOS	REN	Saved	
2015	14.08 (4.7, 28.2)	12.33 (4.1, 24.7)	1.75 (0.6, 3.5)	11.99 (4.0, 23.9)	10.5 (3.5, 21.0)	1.49 (0.5, 2.9)	47.32 (15.8, 94.6)	41.1 (13.7, 82.2)	6.22 (2.1, 12.4)	9.46 (3.2, 18.9)
2020	15.44 (5.2, 30.9)	12.73 (4.2, 25.5)	2.71 (0.9, 5.4)	12.77 (4.3, 25.5)	10.55 (3.5, 21.1)	2.22 (0.7, 4.4)	52 (17.3, 104)	42.45 (14.2, 84.9)	9.55 (3.2, 19.1)	14.48 (4.8, 28.9)
2025	16.03 (5.3, 32.1)	12.59 (4.2, 25.2)	3.44 (1.2, 6.9)	12.84 (4.3, 25.7)	10.08 (3.4, 20.2)	2.77 (0.9, 5.5)	54.02 (18.0, 108)	41.86 (13.9, 83.7)	12.16 (4.1, 24.3)	18.37 (6.1, 36.7)
2030	15.71 (5.2, 31.4)	11.94 (3.9, 23.9)	3.78 (1.3, 7.6)	12.21 (4.1, 24.4)	9.22 (3.1, 18.4)	2.99 (1.0, 5.9)	52.89 (17.6, 106)	39.51 (13.2, 79.1)	13.38 (4.5, 26.8)	20.15 (6.7, 40.3)

^1^ The values in brackets means confident interval (C.I) of 95%. (C.I low 95%, C.I high 95%), the value out of the brackets means the middle value.

**Table 8 ijerph-18-00623-t008:** Work loss days (unit: 10 million days) ^1^.

Year	Beijing		Tianjin		Hebei		Total Saved
	FOS	REN	FOS	REN	FOS	REN	
2015	1.62 (1.4, 1.9)	1.42 (1.2, 1.6)	1.30 (1.1, 1.5)	1.14 (1.0, 1.3)	5.02 (4.3, 5.8)	4.36 (3.7, 5.0)	1.02 (0.9, 1.2)
2020	1.51 (1.3, 1.7)	1.25 (1.1, 1.4)	1.21 (1.0, 1.4)	1.00 (0.9, 1.2)	4.69 (3.9, 5.4)	3.83 (3.3, 4.4)	1.34 (1.1, 1.5)
2025	1.38 (1.2, 1.6)	1.08 (0.9, 1.3)	1.10 (0.9, 1.3)	0.87 (0.7, 1.0)	4.29 (3.6, 4.9)	3.32 (2.8, 3.8)	1.49 (1.3, 1.7)
2030	1.22 (1.0, 1.4)	0.93 (0.8, 1.1)	0.98 (0.8, 1.1)	0.74 (0.6, 0.9)	3.81 (3.2, 4.4)	2.85 (2.4, 3.3)	1.49 (1.3, 1.7)

^1^ The values in brackets means confident interval (C.I) of 95%. (C.I low 95%, C.I high 95%), the value out of the brackets means the median value.

## Data Availability

The data presented in this study are available on request from the corresponding author..

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
