# Peer review of "Impact of the Electric Vehicle Policies on Environment and Health in the Beijing–Tianjin–Hebei Region"

_ijerph, 2021, doi:10.3390/ijerph18020623_

Round 1
Reviewer 1 Report
This study set two scenarios: the electric vehicle policies scenario (REN) and the electric vehicle. policies scenario (FOS) to compare electric vehicle policy's environmental and health impact. It quantified the health benefits of the policies in the BTH region by using an integrated assessment framework. The topic per se is interesting and the authors have done a reasonable job of conducting the empirical work. Although is a good paper, it got some major issues which can be fixed in light of the following suggestions.
- In the Introduction, you need to connect the state of the art to your paper goals. Please follow the literature review by a clear and concise state of the art analysis. This should clearly show the knowledge gaps identified and link them to your paper goals. Please reason both the novelty and the relevance of your paper goals.
- The authors have to provide an extra section with literature review
- The empirical\theoretical contributions of the paper should be clarified in the introduction and in the conclusion.
- It is important to identify the most important result and explain how the result is obtained and why it matters. Please make sure that your main finding is robust and you are able to explain the mechanism of the impact with unambiguous supporting evidence in your paper.
- Results section is very briefly; I think that the analysis in this paper should be more comprehensive
- In the conclusions, in addition to summarising the actions taken and results, please strengthen the explanation of their significance. It is recommended to use quantitative reasoning comparing with appropriate benchmarks, especially those stemming from previous work.
Reviewer 2 Report
This study covers an important and interesting topic.
The paper is well organized and the methods are clearly presented ans seems adequate.
My overall opinion about the paper is very positive.
Despite this fact, the paper has two main limitations in my opinion:
(i) I feel the lack of a (short) literature review, providing the background for the empirical exercise;
(ii) the discussion of the evidence could be stronger, namely through the comparison with other studies.
Reviewer 3 Report
The paper deals with a very interesting and current issue related to "Impact of the electric vehicle policies on environment and health in the Beijing-Tianjin-Hebei Region", which can be considered in line with the scope of this journal. Despite the importance of the topic and the appropriateness of the adopted approach, the paper seems to have several main issues that must be solved, prior to be considered for publishing, such as:
In the section "Introduction" it is necessary to clearly state the main objective of this work, which is not clear in the initial phase of reading the article. In addition, you must add a paragraph at the end of this section with the structure of the article and a corresponding summary of what is expected to be found in each section. On the other hand, there is a wide range of bibliography on health impacts and other tools for assessing impacts that could be used to diversify and broaden the territorial context of the problem, in general, and other studies can be inserted in the references.
Section 2 presents a certain imbalance in the detail and description of the different subsections. Thus, the reviewer understands that points 2.3 and 2.4, related to the GAINS model and IMED / HEL model should be approached with the same level of detail as the points related to point 2.2 - "Energy calculation method". Thus, it is understood that instead of a reference to documents about where it is possible to find information about how the model works, the authors should provide a brief summary of this process and how it will be applied, indicating what type of information enters the model, how it is processed and how the results are obtained.
Also in section 2., subsection 2.2.1, bearing in mind that the different types of vehicles are converted into three types: cars, buses and trucks, why were the "proportion prediction functions" not defined for these three types of vehicles?
In addition, in subsection 2.2.2, table 4 should appear after expression (3) and not subsection 2.2.1., Leaving the doubt whether the values used to obtain the coefficients in Table 4 are from the "National Bureaus of Statistics" or whether they are calculated based on other sources. In fact, this is one of the most sensitive points of the work as it presents the sources with the respective links, but the authors in section 2.2 could be more elucidative and make a synthesis of the data they use with input parameters that they will present in the results, eg, a summary in terms of average, minimum and maximum values, standard deviation over the values of the energy consumption data, etc. If it were not in this section, at least they could present before the tables / maps with the results of section 3.
In section 3. Results, the authors should provide some information / data on the input parameters of the functions and models used to allow a critical analysis of the results that are presented. Otherwise, it is difficult to discuss what is presented. Regarding the maps in Fig. 3 and Fig. 4 it would be important how modelling was done, indicating the model used, the height for which the maps were generated, etc.
Round 2
Reviewer 2 Report
The new version of the paper answers my key concerns. Thus, in my opinion, the paper can be accepted for publication.